# SGD ON RANDOM MIXTURES:
# PRIVATE MACHINE LEARNING UNDER DATA BREACH THREATS

**Kangwook Lee, Kyungmin Lee, Hoon Kim, Changho Suh & Kannan Ramchandran**
School of EE, KAIST & Dept. of EECS, UC Berkeley
`{kw1jjang, atm13579, gnsrla12, chsuh}@kaist.ac.kr, kannanr@eecs.berkeley.edu`

## ABSTRACT

We propose Stochastic Gradient Descent on Random Mixtures (SGDRM) as a simple way of protecting data under data breach threats. We show that SGDRM converges to the globally optimal point for deep neural networks with linear activations while being differentially private. We also train nonlinear neural networks with private mixtures as the training data, proving the practicality of SGDRM.

## 1 INTRODUCTION

In a wide variety of machine learning problems, the training dataset consists of sensitive data such as medical records, personal photos, or proprietary data. In such applications, one needs to train a machine learning model without compromising the privacy (Abadi et al., 2017). Various attack models have been considered in the literature such as black-box attacks and white-box attacks. In this work, we consider a stronger attack scenario, which we dub the 'data-breach attack' model. Under this model, the attacker has access to the input dataset that is being fed to the training algorithm. In order to protect the data under this model, the data owner needs to transform the original dataset into a different form, and provide the training algorithm with the transformed dataset. Further, one also needs a training algorithm that can efficiently learn a model even with the transformed dataset. As our solution, we propose *Stochastic Gradient Descent on Random Mixtures (SGDRM)*, which consists of a data publishing algorithm and a training algorithm. The key idea is simple:

*SGDRM randomly mixes training data points and runs the standard SGD algorithm on them.*

The data-publishing algorithm of our scheme is as follows: the data owner picks a random subset of the entire dataset, and then publishes a random linear combination with additive noise. See Fig. 1 for sample mixtures. Given such random affine combinations of data points, or random mixtures, the learner runs the vanilla SGD algorithm on them. In this work, we theoretically show that 1) random mixtures with additive noise are differentially private, and 2) the SGD converges to the global optimum for linear neural networks. Further, we empirically show that our algorithm works well even for general neural networks such as deep convolutional neural networks (CNNs).

**Related Work:** The 'data-breach attack' model is motivated by the following two applications. The first application is 'machine learning on the cloud'. Consider a data owner who uploads a training dataset to the cloud and run a machine learning algorithm on it. If private access to the cloud is compromised or the cloud infrastructure has some security vulnerability, the entire training data is subject to data breach. Hence, the process of learning on the cloud can be viewed as subject to data-breach attacks. Another scenario is where a learner wants to train a model using crowdsourced private datasets, where the learner can be viewed as a potential data-breach attacker.

Researchers have proposed several solutions to private machine learning (Abadi et al., 2017; Chaudhuri et al., 2011; Abadi et al., 2016; Song et al., 2013). However, most of the existing approaches are not applicable to the data-breach model. A few notable exceptions are recent works on generating private datasets via GAN (Beaulieu-Jones et al., 2017). Another line of related works propose the idea of mixing pairs of data points to augment the training dataset, achieving the state-of-the-art

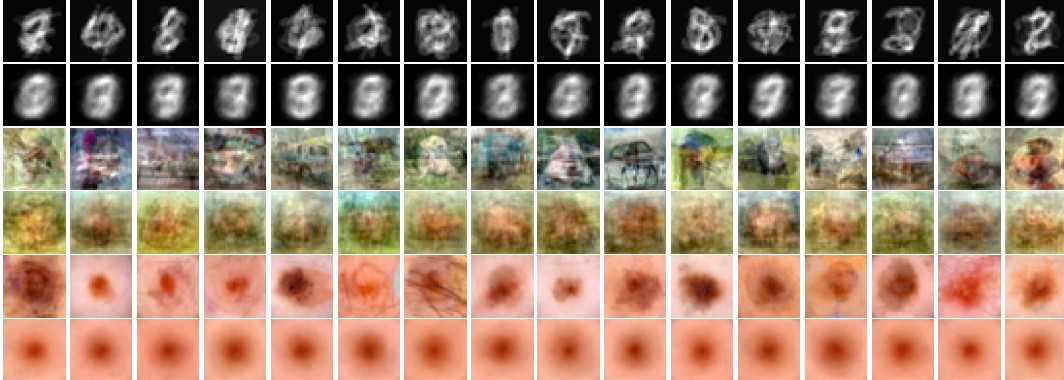

Figure 1: Sample mixtures generated from MNIST, CIFAR10, and Skin Lesion datasets (Codella et al., 2017). The 1st, 3rd, and 5th rows are sample mixtures generated by mixing 8 original images with random coefficients; shown on the other rows are mixtures of 128 images, which look almost random. The one-hot-encoded labels of the images are also mixed with the same mixing coefficients. For instance, the label of the top left corner image is $(0.10, 0.09, 0.12, 0.10, 0.11, 0.09, 0.14, 0.10, 0.06, 0.09)$. Our SGDRM algorithm runs an SGD-like algorithm to train a classification model with these mixtures. We summarize the classification performances in Table 1. For instance, the SGDRM achieves the test accuracy of $0.92$ on the MNIST test dataset when each training data point is a random mixture of 128 MNIST training data points.

performances on some tasks (Zhang et al., 2018; Inoue, 2018). While their algorithms also protect datasets by mixing data points, clear distinctions exist between theirs and SGDRM. First, their algorithms only mix two data points with fixed coefficients, while we mix an arbitrary number of data points with i.i.d. coefficients. Indeed, we show that using i.i.d. random coefficients is a key to proving our convergence guarantees. Also, SGDRM is differentially private while theirs are not.

## 2 PROBLEM FORMULATION AND SGDRM

**Notation:** $[n] := \{1, 2, \ldots, n\}$; $I_n$ is the identity matrix of size $n \times n$; $[A_1; A_2] := [A_1^\top \ A_2^\top]^\top$; $C \sim \text{Bern}(p)$ if $C$ is an i.i.d. Bernoulli random variable with parameter $p$; $C \sim N(\mu, \sigma^2)$ if $C$ is an i.i.d. Gaussian random variable whose mean is $\mu$ and variance is $\sigma^2$; $C \sim \text{Bern}(p) \cdot N(\mu, \sigma^2)$ if $C = C_1 C_2$ for $C_1 \sim \text{Bern}(p)$ and $C_2 \sim N(\mu, \sigma^2)$; the $L_2$ norm of a vector or the Frobenius norm of a matrix is denoted by $\|\cdot\|$; and the minimum singular value of matrix $X$ is denoted by $s_{\min}(X)$.

Consider a data owner who holds a sensitive training dataset and wants to publish a processed dataset such that the published dataset does not leak too much private information about the original dataset. At the same time, the learner needs to run an efficient learning algorithm on the published dataset. Our goal is to jointly design a private data publishing mechanism for the data owner and an efficient training algorithm for the learner to satisfy the above requirements. In this work, we focus on the standard supervised learning setting with $n$ data points. That is, the data owner holds a dataset, which consists of a feature matrix $X = [X_1 \ X_2 \ \cdots \ X_n] \in \mathbb{R}^{d_x \times n}$ and a label matrix $Y = [Y_1 \ Y_2 \ \cdots \ Y_n] \in \mathbb{R}^{d_y \times n}$, where $(X_i, Y_i)$ is the $i^{\text{th}}$ data point. Without loss of generality, we will assume that $\|X_i\| \leq 1$ and $\|Y_i\| \leq 1$ for all $i \in [n]$. The goal of the learner is to find a model, parameterized by $W$, that minimizes a given loss function, say $\mathcal{J}(W, X, Y) = \frac{1}{n} \sum_{i=1}^{n} \mathcal{L}(W, X_i, Y_i)$. Here, we assume that the squared Frobenius norm of the model parameter $W$ is always bounded by a large constant, say $M$. Note that this loss function, which can be viewed as an empirical risk function, depends on the original dataset but the learner does not have an access to it.

We now describe how we quantify the level of privacy and utility. For privacy, we choose *differential privacy* as the privacy notion (Dwork, 2008). (The two datasets $(X, Y)$ and $(X', Y')$ are adjacent if $[X; Y]$ and $[X'; Y']$ differs only in a single column.) For utility, we consider *the optimality gap*, i.e., $\mathbb{E}[\mathcal{J}(\hat{W}) - \mathcal{J}(W^\star)]$, where $W^\star$ is the global minimum of the loss function.

Algorithm 1 is the pseudocode of our SGDRM (SGD on Random Mixtures) algorithm. As we briefly described earlier, our algorithm achieves privacy by mixing randomly chosen data points with random coefficients. We denote the average number of data points included in each mixture by $\ell$, and call it **the mixture width**. Given $T$, the number of data points to be published $T$, the data owner generates $T$ random linear combinations of the data points $[(X_t', Y_t')]_{t \in [T]}$ as $(X_t', Y_t') =$

---

**Algorithm 1** Stochastic Gradient Descent on Random Mixtures
---
1: Learner randomly initializes the model parameter $w_0$
2: **for** $t = 1, \ldots, T$ **do**
3:    Data owner shares a mixture $(X'_t, Y'_t) \leftarrow (XC_t + Q_t, YC_t + R_t)$ with learner
4:    Learner computes the gradient w.r.t. the mixed data point: $g_t \leftarrow \nabla_W \mathcal{L}(W_t, X', Y')$
5:    Learner updates the model parameter: $W_{t+1} = W_t - \alpha_t V(g_t, W_t)$
6: **end for**
7: Learner outputs the model parameter obtained in the last iteration $W_{T+1}$

---

$\left( \sum_{i=1}^{n} C_{t,i} X_i + Q_t, \sum_{i=1}^{n} C_{t,i} Y_i + R_t \right)$, where $C_{t,i} \sim \mathrm{Bern}(\ell/n) \cdot N(0, 1/\ell)$, $Q_t \sim N(0, \sigma^2 I_{d_x})$, and $R_t \sim N(0, \sigma^2 I_{d_y})$. Defining $C_t = [C_{t,1}; C_{t,2}; \ldots; C_{t,n}]$, one can also write $(X'_t, Y'_t) = (XC_t + Q_t, YC_t + R_t)$. Note that our algorithm combines *both* the features and labels.

Our training algorithm is *the standard SGD applied to the random mixtures.*. That is, in iteration $t$, one performs $W_{t+1} = W_t - \alpha_t V(\nabla_W \mathcal{L}(W_t, X'_t, Y'_t), W_t)$, where $V(\cdot)$ removes the bias.

We now present theoretical guarantees: Thm. 1 for privacy and Thm. 2 and Thm. 3 for convergence.

**Theorem 1** (Differential privacy guarantee). *The SGDRM algorithm with the mixture width $\ell$ and the number of iterations $T$ is $(\epsilon, \delta)$ differentially private for $\delta < \frac{1.46\ell}{n}$ if*

$$\sigma^2 \geq \frac{4\sqrt{T}\gamma_{\epsilon,\delta}}{\ell\epsilon^2} \left( \sqrt{T}\gamma_{\epsilon,\delta} \log\left(\frac{4T\ell}{n\delta}\right) \log\left(\frac{2T}{\delta}\right) + 2\epsilon \right), \tag{1}$$

*where $\gamma_{\epsilon,\delta} = \sqrt{2\log(2/\delta) + 8\epsilon} + \sqrt{2\log(2/\delta)}$ and $\sigma^2$ is the variance of $R_t$ and $Q_t$.*

**Theorem 2** (Least squares). *Assume that $\mathcal{J}(W) = \frac{1}{2n}\|WX - Y\|_F^2$ and $W^* = \arg\min_W \mathcal{J}(W)$. Then, the output of SGDRM with $\alpha_t = \frac{n}{s_{min}(X)t}$ and $V(g, W) = g - \sigma^2 W$ satisfies*

$$\mathbb{E}[\mathcal{J}(W_T) - \mathcal{J}(W^*)] \leq \frac{c_\sigma(1 + \log T)}{T}, \tag{2}$$

*where $c_\sigma = c_1 + c_2\sigma^2 + c_3\sigma^4$ and $(c_1, c_2, c_3)$ are positive constants, depending only on $(X, Y, W)$.*

**Theorem 3** (Deep linear neural networks). *Assume that $\mathcal{J}(W) = \frac{1}{2n}\|W_{H+1}W_H \cdots W_2 W_1 X - Y\|_F^2$ for $H \geq 1$. Then, the output of SGDRM with $V(\cdot) = (V_1(\cdot), V_2(\cdot), \ldots, V_{H+1}(\cdot))$, where $V_i(g, W) = g_i - \sigma^2 R_{i+}^T R_{i+} W_i R_{i-} R_{i-}^T$, $g_i = \nabla_{W_i} \mathcal{L}(W, X', Y')$, $R_{i+} = W_{H+1} \cdots W_{i+1}$, $R_{i-} = W_{i-1} \cdots W_1$ satisfies*

$$\lim_{T \to \infty} \mathbb{E}[\|\nabla_W \mathcal{J}(W_T)\|_F^2] \to 0. \tag{3}$$

## 3 EXPERIMENTAL RESULTS

Our convergence guarantee is specific to linear neural networks. To observe its behavior for general neural networks, we run SGDRM for image classification tasks: MNIST, CIFAR10, and Skin Lesion. We generate mixtures with uniform random coefficients with zero noise. Shown in Fig. 1 are sample mixtures: They look completely random to human eyes as $\ell$ increases. For the classification network, we use 2 convolutional layers followed by 3 fully connected layers of size $(100, 100, 10)$. For computational efficiency, instead of the standard SGD, we use the Adam optimizer with minibatches of size 8. Table 1 summarizes the results. Note that these performances are with respect to the original 'unmixed' test dataset. We observe that the performance of SGDRM with $\ell = 2$ slightly outperforms the standard SGD, which is similar to the phenomenon observed in Zhang et al. (2018). After that, as $\ell$ gets larger, the performance starts degrading while becoming more private.

Table 1: Performance of SGDRM on classification tasks.

| Data set/Metric | SGD | $\ell = 2$ | $\ell = 4$ | $\ell = 8$ | $\ell = 16$ | $\ell = 32$ | $\ell = 64$ | $\ell = 128$ |
|---|---|---|---|---|---|---|---|---|
| MNIST/Accuracy | 0.9921 | 0.9919 | 0.9862 | 0.9812 | 0.9687 | 0.9369 | 0.9297 | 0.9201 |
| CIFAR10/Accuracy | 0.7157 | 0.7347 | 0.7190 | 0.6339 | 0.5395 | 0.4296 | 0.3789 | 0.3644 |
| Skin Lesion/AUC | 0.816 | 0.829 | 0.784 | 0.780 | 0.795 | 0.687 | 0.708 | 0.654 |

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
