# OpenReview forum: "SGD on Random Mixtures: Private Machine Learning under Data Breach Threats"
_ICLR.cc/2018/Workshop — Accept_

### Official Review · AnonReviewer2 · 2018-03-10
**The result seems to be theoretically sound, but the setting is not clear**

**Rating:** 7
**Confidence:** 4

**Review:**

The motivation is clear and well understood.
The result seems to be theoretically sound, but the setting is not clear.
The authors mention that the convergence result is specific to linear neural network models. If so, the resulting model is represented by a linear model. If this is correct, the analytic solution is obtainable and it is unclear to me if the results obtained in this study are meaningful. Please clarify this point in the revised version.

---

### Official Review · AnonReviewer1 · 2018-03-10
**Review of "SGD on Random Mixtures"**

**Rating:** 6
**Confidence:** 4

**Review:**

The paper presents an interesting new concept: protection of an ML pipeline against a data breach attack. The authors propose distorting a dataset  to produce an alternate version that (a) is differentially private w.r.t. the original disclosure, (b) can be used to train a model with as close an accuracy as possible. The authors propose a methodology that mixes inputs, by computing new datapoints as follows: each datapoint is a (differentially private) average between $\ell$ inputs randomly selected. $\ell$ establishes a fidelity/diff. privacy tradeoff. The authors provide an accuracy guarantee in the case of what they refer to as "Deep linear neural networks", namely, neural networks with a linear activations. An evaluation over a 5 layer network shows that the method maintains good performance even if the linear activation is amended.

This is, overall, a nice idea, with a nice execution. The paper is formal, precise, and clear, despite the lack of space. The "magic/surprizing" observation here is that training over a mixture has guarantees even when loss is measured with respect to the original dataset, not the transformed dataset.

I have a few concerns. First, there are no "deep linear networks": this is somewhat of an oxymoron. A network with a linear activation function is equivalent to linear regression. There no reason to go "deep" in such a case. Second, the guarantee seems straightforward (but not trivial, due to DP) in the case when the data is generated by a linear model, as linear combinations would still be helpful in training the same linear model. The result does not seem surprising in light of these two observations, and the true challenge, both in terms of a desirable algorithm as well as its analysis, seems to lie in the non-linear case.

---

### Official Review · AnonReviewer3 · 2018-03-11
**Need more work on the privacy part**

**Rating:** 5
**Confidence:** 4

**Review:**

The paper proposes an algorithm called SGDRM that performs SGD for convex and non-convex problems and protects privacy against data-breach attack. SGDRM constructs several random noisy mixtures of the original dataset and performs normal SGD on the mixtures. The paper shows the differential privacy guarantees and convergence guarantee for least square and linear neural network; it also shows experimental results with general neural networks, comparing SGDRM with different mixture widths.

The idea of using random noisy mixtures to protect privacy is quite interesting. It is nice to see that using mixture can sometimes lead to better results than using the original dataset, and helps with privacy as well. However, I'm not fully convinced about the differential privacy part, as the experiments are done with zero noise and thus does not guarantee differential privacy. It would be better if there is some result for what (epsilon, delta) we can get for common machine learning tasks, and how that compares with existing works. Also, to show the significance of using the random mixtures, providing more comparison with Zhang et al. (2018) may help.

Some more comments:
About privacy:
- I think the data-breach attack is similar to local differential privacy, which considers publishing a noisy version of the original dataset. Is that the case? If so, you may want to say something about local dp and other ml algorithms that guarantees local dp in related work.
- If related, it might be better if you mention how random sampling helps with privacy in general, with some intuition and previous works.
- It would be better if you show some values of (epsilon, delta) in real examples.
- It might be better if you show experimental results on the tradeoff between T and l, and how the accuracy/privacy compare with adding noise to the gradients (instead of to the samples).

About the mixture of samples:
- Can you provide some intuition on why mixing samples may help training? Does it help in a certain kind of data / NN? Is it possible for it to help with accuracy for l > 2?

Typo:
- in page 2 second last line "Given T, the number of data points to be published T".

---

### Decision · Program_Chairs · 2018-03-20
**ICLR 2018 Workshop Acceptance Decision**

**Decision:**

Accept

**Comment:**

Congratulations, your paper was accepted to the ICLR workshop.